# HalluField: Detecting LLM Hallucinations via Field-Theoretic Modeling

## Abstract

Large Language Models (LLMs) exhibit impressive reasoning and question-answering capabilities. However, they often produce inaccurate or unreliable content known as hallucinations. This unreliability significantly limits their deployment in high-stakes applications. Thus, there is a growing need for a general-purpose method to detect hallucinations in LLMs. In this work, we introduce HalluField, a novel field-theoretic approach for hallucination detection based on a parametrized variational principle and thermodynamics. Inspired by thermodynamics, HalluField models an LLM's response to a given query and temperature setting as a collection of discrete likelihood token paths, each associated with a corresponding energy and entropy. By analyzing how energy and entropy distributions vary across token paths under changes in temperature and likelihood, HalluField quantifies the semantic stability of a response. Hallucinations are then detected by identifying unstable or erratic behavior in this energy landscape. HalluField is computationally efficient and highly practical: it operates directly on the model's output logits without requiring fine-tuning or auxiliary neural networks. Notably, the method is grounded in a principled physical interpretation, drawing analogies to the first law of thermodynamics. Remarkably, by modeling LLM behavior through this physical lens, HalluField achieves state-of-the-art hallucination detection performance across models and datasets.

## 1 Introduction

Hallucination is a critical and persistent challenge in the use of LLMs. These models, while demonstrating remarkable capabilities across diverse domains, including medicine, education, and software development, are prone to generating outputs that are factually incorrect or logically inconsistent. These hallucinations undermine trust and reliability, especially in high-stakes applications.

Current efforts to mitigate hallucinations in LLMs primarily rely on uncertainty estimation or probabilistic approaches (see Section 2). As LLMs often exhibit well-calibrated predictive confidence, high uncertainty in structured tasks, such as multiple-choice question answering, can serve as a useful proxy for identifying potential hallucinations. However, these approaches face significant limitations. The probabilistic outputs of LLMs are typically high-dimensional, while labeled examples of hallucinations are scarce. This imbalance, often involving only a few thousand hallucinated examples compared to probability vectors with dimensionalities ranging from $10^4$ to $10^5$, makes it extremely challenging to extract meaningful and reliable signals for detection. Particularly, existing methods often resort to coarse statistical measures, such as the log-probability of generating the correct answer given a reference $P_{\text{true}}$ (Kadavath et al., 2022), the entropy of an ensemble of perturbed model outputs on the same query (Farquhar et al., 2024), or related variants (Nikitin et al., 2024). While these signals can be indicative, they capture only a fraction of the model's internal fingerprint. Much of the rich structure in the LLM's response is discarded due to its high complexity, despite its potential to significantly enhance hallucination detection performance.

The goal of this work is to design a theoretical framework to model the response behavior of LLMs in a way that captures the rich information embedded in their output logits. We demonstrate the effectiveness of this framework through its strong performance in hallucination detection. Motivated by principles from thermodynamics, our approach applies parameterized variational principles to model an LLM's response to a given query. Figure 1 provides an intuition of our approach: at a

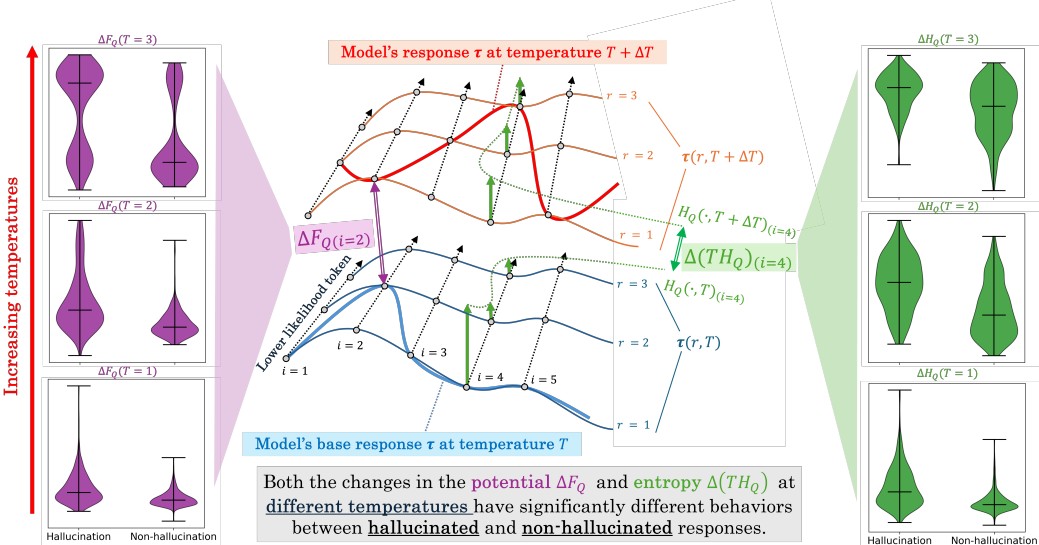

Figure 1: Schematic of the HalluField framework for analyzing LLM responses: The change in free energy, $\Delta\mathbb{F}_Q$, and entropy, $\Delta(T\mathbb{H}_Q)$, between the generated token sequence of interest $\{r_i = r_i^0\}_{i=1}^N$ at the base temperature $T$ (tokens along the *blue* path) and the sequence generated at a perturbed temperature $T + \Delta T$ (tokens along the *red* path) are our proposed signatures for hallucination detection. $\Delta\mathbb{F}_Q$ for $i$th token, formulated in equation 7b, is shown by purple arrows for $i = 2$. Similarly, $\Delta(T\mathbb{H}_Q)$ for $i$th token with entropy defined in equation 8b, is shown by green arrows. Violin-plots of the total $\mathbb{F}_Q$ and $\mathbb{H}_Q$ for hallucinated versus non-hallucinated responses at different temperatures (evaluated on LLaMa-2-7B-Chat in the TriviaQA dataset) illustrate why combining these quantities is a promising approach for hallucination detection.

specified temperature, the response is represented as a collection of discrete token likelihood paths $\tau$, each associated with a corresponding energy $\mathbb{F}_Q$ and entropy $\mathbb{H}_Q$. This formulation enables a principled interpretation and quantification of the structure in LLM outputs, resulting in strong signatures for hallucination detection as illustrated in the violin plots for hallucinated versus non-hallucinated responses in the figure. Another important feature that distinguishes HalluField from state-of-the-art detection methods (Farquhar et al., 2024; Nikitin et al., 2024) is that the computation of energy and entropy does not require the use of auxiliary LLMs. This design not only reduces computational overhead but also avoids the additional uncertainty and potential errors introduced by relying on extra models, which is critical in high-stakes applications.

**Organization.** In Section 2, we provide background and discuss preliminaries and related work, particularly in the context of hallucination and uncertainty in LLMs, as well as classical variational principles. In Section 3, inspired by thermodynamics and variational principles, we develop a theoretical framework for characterizing the responses of LLMs; particularly, we develop notions of energy, entropy, and variations in the LLM setting. This framework allows us to construct the HalluField algorithm for detecting hallucinations, discussed in Section 4. In Section 5, we demonstrate our proposed method on a variety of large language models and datasets, which show excellent performance with respect to Area Under the Curve (AUC), accuracy, and run time.

## 2 PRELIMINARIES AND RELATED WORK

**Autoregressive Generation in LLMs:** An LLM generates textual sequences by autoregressively modeling the conditional probabilities over a discrete token space. Given a query $Q$, the LLM produces an output token sequence $\{\tau_i\}_{i=1}^N$, where each token $\tau_i$ belongs to a finite vocabulary $\mathbb{T}$. The generation process is governed by the joint probability $\prod_{i=1}^N P(\tau_i \mid Q, \tau_{<i})$, where the model recursively estimates the next token conditioned on all previous tokens and the input query. At each step $i$, the LLM outputs a logit vector $\ell_i = f_\theta(Q, \tau_{<i}) \in \mathbb{R}^{|\mathbb{T}|}$, which is transformed into a

probability distribution via the softmax function modulated by a temperature parameter $T \geq 0$:

$$P(\tau_i \mid Q, \tau_{<i}) = \text{softmax}\left(\ell_i / T\right). \tag{1}$$

The temperature $T$ controls the sharpness of the distribution as higher values of $T$ lead to more random outputs. After that, the generated token will be selected by sampling from $P(\tau_i \mid Q, \tau_{<i})$.

Due to the stochastic nature of the generation process, we characterize the LLM's response as a sequence of tokens parameterized by the temperature $T$ and a vector of likelihood ranks $\mathbf{r} \in (\mathbb{Z}^+)^N$. Each component $r_i$ of $\mathbf{r}$ indicates that the $i^{\text{th}}$ token in the sequence corresponds to the $r_i^{\text{th}}$ most likely token under the model's predicted distribution at that step. We denote the resulting sequence by $\boldsymbol{\tau}(\mathbf{r}, T) \in \mathbb{T}^*$. For example, $\boldsymbol{\tau}(\mathbf{1}, T)$ corresponds to the case where the model always selects the most likely token at all steps. However, due to randomness introduced by temperature-scaled sampling, the model may select lower-ranked tokens. We denote a specific sequence generated by the model at temperature $T$ as $\boldsymbol{\tau}(\mathbf{r}^0, T)$, where $\mathbf{r}^0$ records the rank of each selected token. The LLM's response that we aim to evaluate for hallucination is referred to as the *base response* $\boldsymbol{\tau}(\mathbf{r}^0, T^0)$, defined as the specific output generated at the *base temperature* $T^0$.

**Hallucination and Uncertainty in LLMs:** Hallucination has become an active area of research, with several detection methods proposed to quantify or mitigate model uncertainty. Early work, such as (Kadavath et al., 2022), introduced the $P_{\text{true}}$ measure, which leverages model probabilities as an indicator of correctness. More recently, Farquhar et al. (2024) proposed *Semantic Entropy* (SE), which measures uncertainty by clustering semantically equivalent generations and computing entropy over these clusters. Given a query $Q$, the SE of $Q$, denoted by $SE_Q$ is given by:

$$SE_Q = -\sum_{C \in \Omega} P(C \mid Q) \log P(C \mid Q) = -\sum_{C \in \Omega} \left(\sum_{s \in C} P(s \mid Q)\right) \log \left(\sum_{s \in C} P(s \mid Q)\right), \tag{2}$$

where $\Omega$ is the set of all semantic clusters and $s$ is the model's response. In practice, SE estimates $\Omega$ and $SE_Q$ by querying another LLM and employing a Rao-Blackwellized Monte Carlo estimator, i.e., $P'(C_i \mid Q) \approx P(C_i \mid Q)/(\sum_j P(C_j \mid Q))$, respectively. Extending this line of work, Nikitin et al. (2024) introduced *Kernel Language Entropy* (KLE), which estimates fine-grained semantic uncertainty using kernel methods, enabling improved semantic clustering.

Additionally, a variety of different approaches have been explored to detect hallucinations in LLMs, including leveraging external knowledge to validate generated content (Li et al., 2023; Feldman et al., 2023), probing and intervening on hidden states (Burns et al., 2022; Li et al., 2024; Liu et al., 2023), and applying fine-tuning strategies (Kang et al., 2024). A broad body of work has studied uncertainty quantification in sequential and generative models, with many approaches eliciting uncertainty from LLMs via fine-tuning or prompting with prior generations (Kadavath et al., 2022; Chen & Mueller, 2023; Mielke et al., 2022; Lin et al., 2022; Maynez et al., 2020; Ganguli et al., 2023; Ren et al., 2023; Tian et al., 2023; Cohen et al., 2023; Xiao & Wang, 2021; Kuhn et al., 2023). Our HalluField approach is complementary to these directions and provides an alternative perspective on hallucination detection from a variational perspective.

**Variational principles:** Variational principles describe the state of a system as a stationary point of a functional. Variational principles are a well-studied subject and appear in many contexts and applications, such as minimal surfaces (Ulrich Dierkes, 2010), partial differential equations (Marsden & Hughes, 1983), optimization and control (Bloch, 2015; de León et al., 2007), and structure-preserving numerical methods (Marsden & West, 2001). We only provide a brief discussion here relevant for our purposes; for a more detailed discussion, see the classic texts (Abraham & Marsden, 1987; Marsden & Ratiu, 1999; Arnold, 1978) as well as (Tran & Leok, 2025) for a recent review.

Consider an action functional $\mathbb{A} : \mathcal{C} \to \mathbb{R}$, whose domain is a space of curves $\mathcal{C}$, defined by integrating a *density* $A$ along the curve,

$$\mathbb{A}[\mathbf{c}] = \int A(\mathbf{c}) \, ds, \tag{3}$$

where $ds$ is the arclength measure along the curve $\mathbf{c}$. The classical variational principle seeks to find an extremal curve $\mathbf{c}$ which is stationary with respect to variations $\delta \mathbf{c}$ of the action:

$$0 = \Delta \mathbb{A}[\mathbf{c}] = \int \frac{\delta A(\mathbf{c})}{\delta \mathbf{c}} ds, \quad \text{where} \quad \frac{\delta A(\mathbf{c})}{\delta \mathbf{c}} := \frac{\partial A(\mathbf{c})}{\partial \mathbf{c}} \cdot \delta \mathbf{c}. \tag{4}$$

The classical equations of motion seek a stationary point of the function $A$, i.e., $\partial A/\partial \mathbf{c} = 0$.

# 3 VARIATIONAL FRAMEWORK TO STUDY RESPONSES OF LLMS

Our fundamental quantity for detecting LLM hallucinations will be the change in the *internal energy functional* on the space of sequences of tokens. According to the first law of thermodynamics (Landau & Lifshitz, 1980), the *total variation* in the internal energy functional $\mathbb{U}$ is given by

$$\delta\mathbb{U} = T\delta\mathbb{H} + W, \tag{5}$$

where $T$ is the temperature, $\mathbb{H}$ is the entropy functional, and $W$ is the work done on the system.

In thermodynamics, a system is considered stable if small changes in its state do not cause it to spontaneously move to a different state. In terms of internal energy, stability is closely related to local minima in $\mathbb{U}$, while sharp increases in $\mathbb{U}$ may indicate unstable, short-lived configurations (Landau & Lifshitz, 1980). We hypothesize that hallucinated output corresponds to high-energy and less coherent configurations in the token space. Monitoring $\delta\mathbb{U}$ across temperatures may thus help identify unreliable responses: *For a hallucinated response whose internal energy is already high, increasing its temperature has little effect on its internal energy, resulting in a low total variation $\delta\mathbb{U}$. In contrast, raising the temperature would change a correct, low-energy response into an incorrect, high-energy one, leading to a high $\delta\mathbb{U}$.* While this hypothesis is supported by our experimental results in Section 5, the violin plots in Figure 1, and the results in Figure 2, additional supporting evidence is provided in more detail in Appendix A.

Our approach aligns with entropy-based methods such as KLE and SE, which associate high uncertainty with untrustworthy behavior, but it differs in important respects, as will be shown. First, it leverages thermodynamics to systematically compute hallucination signatures across temperatures using the total variation. Second, it captures both entropy and free energy, an intrinsic measure of the reliability of the base response itself, offering a more robust detection signal. Third, it operates directly on response trajectories without relying on auxiliary LLMs, reducing overhead and error.

Nevertheless, directly applying equation 5 to token sequences is intractable as it requires treating entropy as the independent variable and expressing the remaining quantities (temperature and work) as functions of the entropy. In contrast, for LLMs, it is much more convenient to parameterize these quantities in terms of temperature, since temperature can be explicitly controlled by the user. For that purpose, we take the Legendre transform of $\mathbb{U}$, given by the free energy functional (Landau & Lifshitz, 1980): $\mathbb{F} = \mathbb{U} - T\mathbb{H}$. Crucially, the free energy functional is now a function of the temperature, which will allow us to construct a formula to calculate its variation. Subsequently, the quantity of interest $\delta\mathbb{U}$ can be expressed as

$$\delta\mathbb{U} = \delta\mathbb{F} + \delta(T\mathbb{H}). \tag{6}$$

Evaluating equation 6 for LLM's responses still requires several key ingredients: the free energy functional $\mathbb{F}$, the entropy functional $\mathbb{H}$, and a framework to compute the total variations $\delta$ of functionals defined on the space of token sequences as the temperature varies.

**Free energy functional $\mathbb{F}$:** In physical systems, the free energy functional $\mathbb{F}$ represents the *useful* portion of energy, i.e., the amount of work that can be extracted under constraints of temperature and entropy. In the LLM setting, $\mathbb{F}$ can be considered as a measure of sequence coherence and confidence, based on the conditional probabilities of generated tokens. Analogous to how free energy is defined over the probability distribution of physical microstates in statistical thermodynamics, we formulate $\mathbb{F}$ for LLMs as a scalar functional over the conditional probabilities of the output token sequence $\boldsymbol{\tau}$. This formulation satisfies key properties such as *linearity*, *monotonicity*, and *positivity* (Landau & Lifshitz, 1980), leading to the following form (details in Appendix B.2):

$$\mathbb{F}_Q(\boldsymbol{\tau}(\mathbf{r}, T)) = \sum_{i=1}^{N} F_Q(\tau_i(r_i, T)), \tag{7a}$$

$$F_Q(\tau_i(r_i, T)) = -\log P(\tau_i(r_i, T)|\{\tau_j(r_j, T)\}_{j=1}^{i-1}, Q), \tag{7b}$$

where the probability in equation 7b is the parameterized version of the probability in equation 1.

**Entropy functional $\mathbb{H}$:** On the other hand, $\mathbb{H}$ captures the uncertainty of the generated token sequences (details in Appendix B.3). High entropy corresponds to a broad distribution over plausible next tokens, which encourages diversity but also increases the risk of hallucinated content.

$$\mathbb{H}_Q(\boldsymbol{\tau}(\cdot, T)) = \sum_{i=1}^{N} H_Q(\tau_i(\cdot, T)), \tag{8a}$$

$$H_Q(\tau_i(\cdot, T)) = -\sum_{r=1}^{|\mathbb{T}|} P(\tau_i(r, T)|\{\tau_j\}_{j=1}^{i-1}, Q) \times \log P(\tau_i(r, T)|\{\tau_j\}_{j=1}^{i-1}, Q). \tag{8b}$$

Note that, different from SE and related methods (equation 2), our entropy formulation is computed directly at the token level (not the response's level) and does not require an auxiliary LLM.

**Total variation $\delta$:** The total variation describes how to aggregate the variations along the varied parameters. Viewed through the lens of variational principles, we treat both the free energy and the entropy as action functionals on sequences of tokens, offering a unified way to analyze LLM dynamics. Particularly, to compute the total variations $\delta\mathbb{F}$ and $\delta(T\mathbb{H})$ in equation 6, we define the *total variation* of a functional $\delta\mathbb{A}$ as a weighted sum over several different variations of that functional $\Delta\mathbb{A}$. Using temperature as the varied parameter, we heuristically define

$$\delta\mathbb{A}[\boldsymbol{\tau}] := \sum\nolimits_{\Delta T} w(T, \Delta T)\Delta\mathbb{A}[\boldsymbol{\tau}; \Delta T], \qquad (9)$$

where $w$ is a weight function. Intuitively, the total variation of a functional is a weighted response of the functional to several perturbations in the parameter (in this case, temperature). As opposed to only using a single variation, the total variation provides more complete information on the response of a functional to changes in the external parameter; in the context of hallucination detection, the total variation provides a better picture of how LLM responses, characterized by the free energy and entropy functionals, vary as temperature changes.

The details of the construction of the variation $\delta\mathbb{A}$ are provided in Appendix B.1, where we develop a parametrized discrete variation focusing on variations with respect to LLM's temperature. Combined with $\mathbb{F}$ and $\mathbb{H}$, these ingredients enable the explicit computation of equation 6, as will be described in Section 4. The remaining of this section summarizes the correspondence between the proposed quantities in the continuous (conventional), discrete (token space), and parametrized discrete variational principles, which highlights our approach in formulating the total variation $\delta$.

Table 1 shows the correspondence among those variational principles for an arbitrary functional $\mathbb{A}$, which can either be the free energy or the temperature-entropy functional. In the continuous case, trajectories are described by smooth curves, and the action functional $\mathbb{A}[\mathbf{c}]$ is obtained by integration over such trajectories. When passing to the discrete setting, the trajectory becomes a sequence of discrete tokens $\boldsymbol{\tau}$, and the integral is replaced by a finite sum, leading to a discrete action $\mathbb{A}_Q[\boldsymbol{\tau}]$. In the parameterized discrete formulation, the sequence explicitly depends on an external parameter, the temperature $T$. Accordingly, variations are taken with respect to changes in this parameter, resulting in temperature-dependent difference quotients. Additionally, we parameterize the sequence by the likelihood rank $\mathbf{r}$, since this information is also provided by the LLM and will be useful for computing our signatures. Thus, the continuous calculus of variations has direct analogues in discrete token sequences, and further generalizes to parameterized variations that capture model dynamics under external controls such as temperature.

Table 1: Summary of the correspondence among proposed quantities in the continuous, discrete, and parametrized discrete variations of a functional.

| Continuous | Discrete | Parametrized discrete |
|:---:|:---:|:---:|
| $\mathbf{c}$ | $\boldsymbol{\tau}$ | $\boldsymbol{\tau}(\mathbf{r}, T)$ |
| $\mathbb{A}[\mathbf{c}]$ | $\mathbb{A}_Q[\boldsymbol{\tau}]$ | $\mathbb{A}_Q[\boldsymbol{\tau}(\mathbf{r}, T)]$ |
| $\Delta\mathbb{A}[\mathbf{c}]$ | $\Delta\mathbb{A}_Q[\boldsymbol{\tau}; \boldsymbol{\chi}]$ | $\Delta\mathbb{A}_Q[\boldsymbol{\tau}; \Delta T]$ |
| $\int$ | $\sum_{i=1}^{N}$ | $\sum_{i=1}^{N}$ |
| $\frac{\delta A(\mathbf{c})}{\delta \mathbf{c}}$ | $\frac{A_Q(\tau_i) - A_Q(\chi_i)}{d(\tau_i, \chi_i)}$ | $A_Q(\tau_i(r_i, T + \Delta T)) - A_Q(\tau_i(r_i, T))$ |
| $ds$ | $\Delta_i$ | $\Delta_i$ |

# 4 HALLUFIELD ALGORITHM

Given the theoretical framework established in Section 3 and Appendix B, we now describe our HalluField algorithm for hallucination detection. In particular, we describe how to compute the total variation of the free energy $\delta\mathbb{F}_Q$ and the total variation of the temperature-entropy functional $\delta(T\mathbb{H}_Q)$ as a weighted sum over several variations.

**Implementation of the free energy total variation $\delta\mathbb{F}_Q$:** We derive a theoretical form of the free energy variation in equation 30 (Appendix B.2). However, as discussed in more detail in the appendix, the theoretical form is not computable in practice since it requires observing the same LLM's

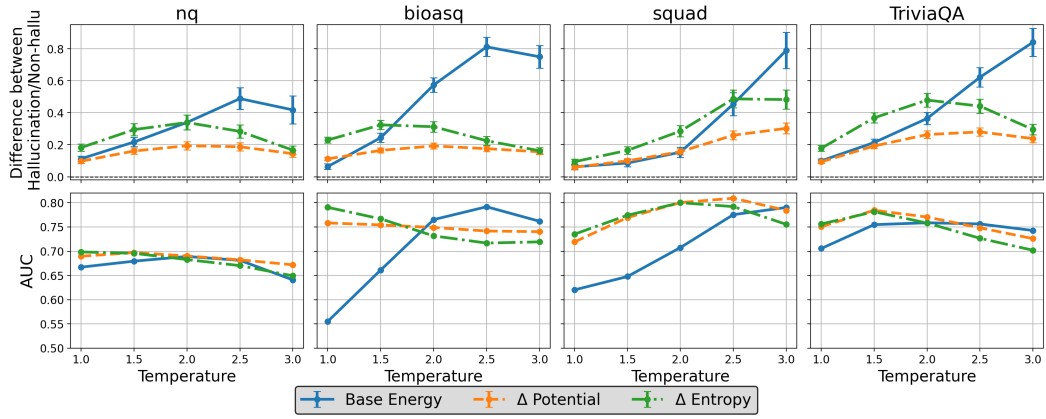

Figure 2: Differences (top) and the AUCs (bottom) of the base energy variation $\Delta\mathbb{B}_Q$, the change in potential $\Delta\mathbb{P}_Q$, and the change in entropy $\Delta(T\mathbb{H}_Q)$ between hallucinated and non-hallucinated responses as a function of temperature for different datasets in LLaMa-2-7B-Chat.

response at higher temperatures. Thus, we decompose the variation into two computable terms: the base variation $\Delta\mathbb{B}_Q$ and the change in potential $\Delta\mathbb{P}_Q$.

Intuitively, the base energy variation $\Delta\mathbb{B}_Q$ captures the change of the free energy in the base response $\boldsymbol{\tau}^0 := \boldsymbol{\tau}(\mathbf{r}^0, T^0)$ versus its energy when the temperature is increased:

$$\Delta\mathbb{B}_Q[\boldsymbol{\tau}^0; \Delta T] = \Delta\mathbb{F}_Q[\boldsymbol{\tau}^0; \Delta T] = \mathbb{F}_Q[\boldsymbol{\tau}(\mathbf{r}^0, T^0 + \Delta T)] - \mathbb{F}_Q[\boldsymbol{\tau}(\mathbf{r}^0, T^0)] \tag{10}$$

However, when the temperature increment $\Delta T$ is too large, the sequence $\mathbf{r}^0$ may never be chosen as the generated tokens. This prevents us from observing $\boldsymbol{\tau}(\mathbf{r}^0, T^0 + \Delta T)$ and computing $\mathbb{F}_Q[\boldsymbol{\tau}(\mathbf{r}^0, T^0 + \Delta T)]$. In such cases, we replace the first term of $\Delta\mathbb{B}_Q$ with the average free energy of the paths generated at that temperature. This yields an approximation of $\Delta\mathbb{F}_Q$:

$$\Delta\mathbb{B}_Q[\boldsymbol{\tau}^0; \Delta T] = \mathbb{E}_\mathbf{r}\left[\mathbb{F}_Q[\boldsymbol{\tau}(\mathbf{r}, T^0 + \Delta T)]\right] - \mathbb{F}_Q[\boldsymbol{\tau}(\mathbf{r}^0, T^0)] \approx \Delta\mathbb{F}_Q[\boldsymbol{\tau}^0; \Delta T]. \tag{11}$$

On the other hand, the change in potential $\Delta\mathbb{P}_Q$ captures the variation in potential that arises when the model generates a different sequence of tokens as a result of the increased temperature:

$$\Delta\mathbb{P}_Q[\boldsymbol{\tau}^0; \Delta T] = \mathbb{E}_\mathbf{r}\left[\mathbf{I}(\mathbf{r} \neq \mathbf{r}^0)\mathbb{F}_Q[\boldsymbol{\tau}(\mathbf{r}, T^0 + \Delta T)] - \mathbb{F}_Q[\boldsymbol{\tau}(\mathbf{r}^0, T^0)]\right], \tag{12}$$

where $\mathrm{I}(\cdot)$ denotes the indicator function (1 if the condition is true, 0 otherwise). As noted earlier (see equation 9), the total variation in free energy, $\delta\mathbb{F}_Q$, is defined as a weighted sum of the base variation $\Delta\mathbb{B}_Q$ and the potential change $\Delta\mathbb{P}_Q$ across multiple $\Delta T$ values, offering a better measure of how the free energy functional varies as opposed to using a single $\Delta T$:

$$\delta\mathbb{F}_Q := \sum_{\Delta T=\Delta T_1}^{\Delta T_n} w_\mathbb{B}(T^0; \Delta T)\Delta\mathbb{B}_Q[\boldsymbol{\tau}^0; \Delta T] + \sum_{\Delta T=\Delta T_1}^{\Delta T_n} w_\mathbb{P}(T^0; \Delta T)\Delta\mathbb{P}_Q[\boldsymbol{\tau}^0; \Delta T]. \tag{13}$$

Figure 2 shows the advantage of using weighted sums across variations of the free energy for hallucination detection. At lower temperatures, the potential change yields a stronger detection capability, as reflected by the larger statistical gap in the signatures and higher AUC values. Conversely, base variation becomes more effective at higher temperatures. Our implementation of HalluField employs the following weighting scheme to encourage the effect of potential changes at low temperatures while maintaining the influence of baseline variation at high temperatures:

$$w_\mathbb{B}(T^0; \Delta T) = T^0 + \Delta T \quad \text{and} \quad w_\mathbb{P}(T^0; \Delta T) = 1/(T^0 + \Delta T)^2 \tag{14}$$

**Implementation of the temperature-entropy total variation** $\delta(T\mathbb{H}_Q)$**:** Similarly, the temperature-entropy functional variation $\Delta(T\mathbb{H}_Q)$ is also measured only when the model generates a different sequence of tokens caused by a higher entropy:

$$\Delta(T\mathbb{H}_Q)[\boldsymbol{\tau}(\cdot, T^0); \Delta T] = T^0\mathbb{E}_\mathbf{r}\left[\mathbf{I}(\mathbf{r} \neq \mathbf{r}^0)\mathbb{H}_Q(\boldsymbol{\tau}(\cdot, T^0 + \Delta T)) - \mathbb{H}_Q(\boldsymbol{\tau}(\cdot, T^0))\right], \tag{15}$$

As in equation 9, the total variation in the temperature-entropy functional is given by a weighted sum over several variations $\Delta T$, where the impact of different temperatures is also depicted in Figure 2:

$$\delta(T\mathbb{H}_Q) := \sum_{\Delta T=\Delta T_1}^{\Delta T_n} w_{T\mathbb{H}}(T^0; \Delta T)\Delta(T\mathbb{H}_Q)[\boldsymbol{\tau}(\cdot, T^0); \Delta T], \qquad (16)$$
$$\text{where } w_{T\mathbb{H}}(T^0; \Delta T) = 1/(T^0 + \Delta T)^2.$$

**HalluField:** Altogether, HalluField computes the change in the internal energy equation 6 by

$$\delta\mathbb{U}_Q = \delta\mathbb{F}_Q + \delta(T\mathbb{H}_Q)$$
$$= \sum_{\Delta T=\Delta T_1}^{\Delta T_n} \left[ (T^0 + \Delta T)\Delta\mathbb{B}_Q[\boldsymbol{\tau}^0; \Delta T] + \frac{\Delta\mathbb{P}_Q[\boldsymbol{\tau}^0; \Delta T]}{(T^0 + \Delta T)^2} + \frac{\Delta(T\mathbb{H}_Q)[\boldsymbol{\tau}(\cdot, T^0); \Delta T]}{(T^0 + \Delta T)^2} \right]. \quad (17)$$

As a final refinement, we incorporate the semantic entropy term $SE_Q$ (equation 2), which leads to the HalluFieldSE algorithm:
$$\text{HalluFieldSE} = \delta\mathbb{U}_Q + \lambda SE_Q,$$
where $\lambda > 0$ is a hyper-parameter, set to 2 in our implementation. The reason that combining $SE_Q$ with $\delta\mathbb{U}_Q$ yields better performance (see Section 5) than using either individually is that they are complementary: $SE_Q$ provides high-level semantic information about the uncertainty of the answer, whereas $\delta\mathbb{U}_Q$ captures detailed low-level information directly from the logits of the response tokens.

The signatures $\delta\mathbb{U}_Q$ and HalluFieldSE are used as predictors for hallucinations. The resulting methods are referred to as HalluField and HalluFieldSE, respectively. The pseudocode of HalluField is provided in Algorithm 1.

---

**Algorithm 1** HalluField

**Input:** The LLM, query $Q$, base temperature $T^0$, number of perturbations $L$, temperature variations $\{\Delta T_1, \ldots, \Delta T_N\}$

**Output:** Total internal energy variation $\delta\mathbb{U}$

1: Ask $Q$ to the LLM with base temperature $T^0$; collect the token probability $P(\tau_i|\{\tau_j\}_{j=1}^{i-1}, Q)$
2: $\delta\mathbb{F}_Q = 0$; $\delta(T\mathbb{H}_Q) = 0$
3: **for** variation $\Delta T$ from $\Delta T_1$ to $\Delta T_n$ **do**
4:     Ask $Q$ to the LLM with temperature $T^0 + \Delta T_i$; collect the token probability
5:     $\delta F_Q \mathrel{+}= w_{\mathbb{B}}(T^0; \Delta T_i)\Delta\mathbb{B}_Q[T^0; \Delta T_i]$     {Use equation 10, equation 11 and equation 7a}
6:     $\delta F_Q \mathrel{+}= w_{\mathbb{P}}(T^0; \Delta T_i)\Delta\mathbb{P}_Q[T; \Delta T_i]$             {Use equation 12 and equation 7a}
7:     $\delta(T\mathbb{H}_Q) \mathrel{+}= w_{T\mathbb{H}}(T^0; \Delta T_i)\Delta(T\mathbb{H}_Q)[T^0; \Delta T_i]$     {Use equation 15 and equation 8a}
8: **end for**
9: **return** $\delta\mathbb{U}_Q = \delta\mathbb{F}_Q + \delta(T\mathbb{H})_Q$

---

## 5 EXPERIMENTAL RESULTS

In this section, we present our experimental results, which demonstrate the strong detection capability of the proposed methods (Tables 2, 3, 4, 5). In addition, we report a substantial improvement in runtime efficiency, achieved by eliminating the need for auxiliary LLM usage (Table 6).

**Experimental settings:** We follow Farquhar et al. (2024) and evaluate our methods on four open-domain question answering datasets: squad, TriviaQA, Natural Questions (nq), and bioasq. squad (Rajpurkar et al., 2016) (Stanford Question Answering Dataset) is a reading comprehension benchmark consisting of over 100,000 crowd-sourced questions on Wikipedia articles, where answers are spans in the provided context. TriviaQA (Joshi et al., 2017) is a large-scale dataset with over 650K question-answer pairs, collected from trivia websites and accompanied by evidence documents. nq (Kwiatkowski et al., 2019) contains real anonymized Google search queries, each paired with a Wikipedia page as context. It includes both short and long answers, making it a challenging and realistic QA benchmark. Finally, bioasq (Krithara et al., 2023) is a manually curated corpus designed for biomedical question answering, built as part of the BioASQ challenge.

Our experiments are conducted on a range of recent open-source LLMs, including the LLaMA-2 (7B, 7B-Chat, and 13B-Chat) (Touvron et al., 2023) and the LLaMA-3.2 variants (1B, 1B-Instruct,

and 3B) (Meta AI, 2024). We also incorporate models from other leading research: Phi-3 Mini-Instruct (Abdin et al., 2023), trained with a heavy emphasis on textbook-style data; Mistral-7B-Instruct (Jiang et al., 2023), a dense transformer optimized for efficiency and instruction-following, and Falcon-7B-Instruct (Penedo et al., 2023), a model trained on high-quality, curated web corpora. More details of the experiments are provided in Appendix C. Our source code is presently under review by our organization, with release expected by the ICLR decision date.

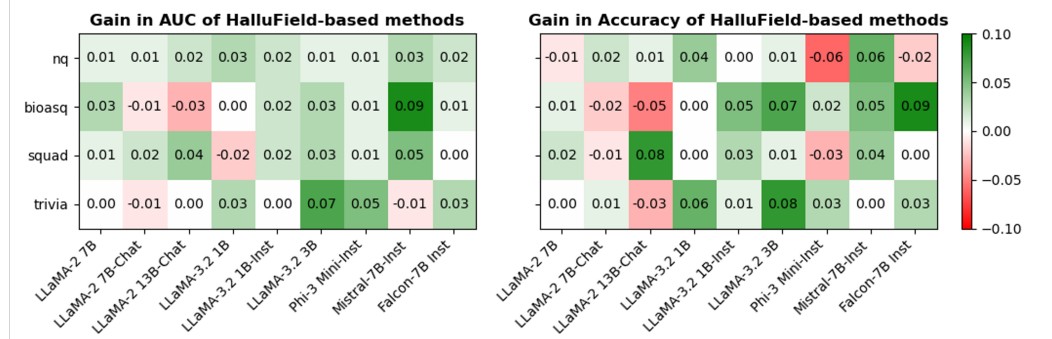

Figure 3: Summary of comparison between our HalluField-based methods (HalluField and Hallu-FieldSE) and Semantic-based methods (KLE and SE) among different models and datasets.

Table 2: AUC/Accuracy on **nq** dataset.

| Model | HalluFieldSE | HalluField | KLE | SE | CE | RE | P(True) |
|---|---|---|---|---|---|---|---|
| LLaMA-2 7B | **0.76** / 0.68 | 0.72 / 0.64 | 0.75 / **0.69** | 0.75 / 0.68 | 0.75 / 0.67 | 0.73 / 0.68 | 0.52 / 0.56 |
| LLaMA-2 7B-Chat | **0.71** / 0.62 | 0.70 / **0.65** | 0.70 / 0.63 | 0.69 / 0.63 | 0.70 / 0.64 | 0.69 / 0.61 | 0.55 / 0.49 |
| LLaMA-2 13B-Chat | **0.76** / 0.68 | 0.75 / 0.67 | 0.74 / 0.67 | 0.74 / 0.67 | 0.74 / **0.68** | 0.71 / 0.67 | 0.59 / 0.63 |
| LLaMA-3.2 1B | **0.72** / 0.58 | 0.71 / **0.69** | 0.69 / 0.64 | 0.69 / 0.65 | 0.69 / 0.65 | 0.70 / 0.67 | 0.51 / 0.32 |
| LLaMA-3.2 1B-Inst | **0.76** / 0.63 | 0.74 / **0.69** | 0.74 / 0.67 | 0.74 / **0.69** | 0.74 / **0.69** | 0.75 / 0.68 | 0.52 / 0.48 |
| LLaMA-3.2 3B | 0.75 / **0.71** | 0.71 / 0.65 | 0.74 / 0.70 | 0.74 / 0.68 | 0.74 / 0.68 | 0.73 / 0.66 | 0.53 / 0.46 |
| Phi-3 Mini-Inst | **0.80** / 0.70 | 0.77 / 0.71 | 0.79 / **0.77** | 0.78 / 0.71 | 0.79 / 0.75 | 0.76 / 0.69 | 0.57 / 0.66 |
| Mistral-7B-Inst | **0.76** / 0.73 | **0.76** / 0.67 | 0.73 / 0.66 | 0.73 / 0.67 | 0.73 / **0.73** | 0.72 / 0.68 | 0.57 / 0.50 |
| Falcon-7B Inst | **0.77** / 0.66 | **0.77** / 0.70 | 0.75 / 0.70 | 0.75 / **0.72** | 0.76 / 0.69 | 0.73 / 0.66 | 0.50 / 0.46 |

Table 3: AUC/Accuracy on **bioasq** dataset.

| Model | HalluFieldSE | HalluField | KLE | SE | CE | RE | P(True) |
|---|---|---|---|---|---|---|---|
| LLaMA-2 7B | **0.83** / 0.73 | 0.80 / 0.70 | 0.80 / 0.72 | 0.80 / 0.67 | 0.80 / 0.66 | 0.73 / **0.74** | 0.66 / 0.54 |
| LLaMA-2 7B-Chat | 0.82 / 0.75 | 0.78 / 0.67 | **0.83** / **0.77** | 0.82 / 0.75 | **0.83** / 0.75 | 0.63 / 0.55 | 0.54 / 0.55 |
| LLaMA-2 13B-Chat | 0.72 / 0.63 | 0.63 / 0.58 | **0.75** / **0.68** | 0.73 / 0.65 | 0.74 / 0.67 | 0.52 / 0.45 | 0.55 / 0.55 |
| LLaMA-3.2 1B | **0.85** / **0.78** | 0.82 / 0.77 | **0.85** / **0.78** | **0.85** / **0.78** | **0.85** / **0.78** | 0.78 / 0.72 | 0.58 / 0.56 |
| LLaMA-3.2 1B-Inst | **0.78** / 0.72 | 0.74 / **0.75** | 0.76 / 0.68 | 0.76 / 0.70 | 0.75 / 0.69 | 0.64 / 0.43 | 0.61 / 0.70 |
| LLaMA-3.2 3B | **0.80** / 0.76 | 0.78 / 0.72 | 0.77 / 0.69 | 0.76 / 0.68 | 0.77 / 0.67 | 0.72 / 0.64 | 0.61 / 0.60 |
| Phi-3 Mini-Inst | **0.80** / 0.75 | 0.79 / 0.73 | 0.79 / 0.73 | 0.78 / 0.73 | 0.79 / 0.74 | 0.72 / 0.68 | 0.52 / 0.53 |
| Mistral-7B-Inst | 0.75 / **0.73** | **0.78** / **0.73** | 0.69 / 0.67 | 0.68 / 0.68 | 0.66 / 0.51 | 0.74 / 0.70 | 0.54 / 0.38 |
| Falcon-7B Inst | **0.81** / 0.74 | 0.78 / **0.82** | 0.78 / 0.73 | 0.80 / 0.69 | 0.79 / 0.72 | 0.64 / 0.61 | 0.59 / 0.61 |

**Detection results:** Figure 3 summarizes the performance of our HalluField-based methods (HalluField and HalluFieldSE) against semantic-based approaches (KLE and SE) across models and datasets. Overall, HalluField consistently achieves competitive performance, highlighting the advantage of integrating structured potential and energy-based signals for hallucination detection.

Besides $P_{\text{true}}$, SE, KLE (see Section 2), we also evaluate HalluField against Regular Entropy (RE) and Cluster-assignment Entropy (CE), which are simplified variants of SE (Farquhar et al., 2024). More detailed results across datasets and models are reported in Table 2, 3, 4 and 5. The results reveal several consistent patterns. First, HalluFieldSE emerges as the strongest overall signal for hallucination detection, frequently achieving the highest AUC values across benchmarks. This suggests that combining semantic evidence with hallucination-oriented features provides a more robust discriminator than relying on any single cue. Closely related, the base HalluField method also performs competitively, often yielding the best accuracy, indicating that its simpler approach relying solely on $\delta\mathbb{U}$ remains effective. KLE and SE tend to trail slightly behind, but remain stable baselines with consistent mid-to-high performance.

Table 4: AUC/Accuracy on **squad** dataset.

| Model | HalluFieldSE | HalluField | KLE | SE | CE | RE | P(True) |
|---|---|---|---|---|---|---|---|
| LLaMA-2 7B | **0.84** / 0.77 | 0.82 / **0.83** | 0.83 / 0.81 | 0.82 / 0.74 | 0.82 / 0.69 | 0.75 / 0.70 | 0.54 / 0.53 |
| LLaMA-2 7B-Chat | **0.82** / 0.74 | 0.74 / 0.75 | 0.79 / **0.76** | 0.80 / 0.75 | 0.75 / 0.74 | 0.70 / 0.73 | 0.56 / 0.57 |
| LLaMA-2 13B-Chat | **0.80** / 0.76 | 0.72 / **0.83** | 0.76 / 0.73 | 0.74 / 0.75 | 0.75 / 0.79 | 0.60 / 0.72 | 0.66 / 0.69 |
| LLaMA-3.2 1B | 0.76 / **0.74** | 0.73 / 0.72 | **0.78** / **0.74** | 0.77 / 0.72 | 0.76 / 0.73 | 0.74 / 0.66 | 0.53 / 0.56 |
| LLaMA-3.2 1B-Inst | **0.83** / **0.79** | 0.81 / **0.79** | 0.81 / 0.75 | 0.81 / 0.76 | 0.81 / 0.74 | 0.72 / 0.71 | 0.60 / 0.58 |
| LLaMA-3.2 3B | **0.82** / **0.79** | 0.80 / 0.67 | 0.79 / 0.74 | 0.78 / 0.78 | 0.78 / 0.74 | 0.73 / 0.64 | 0.54 / 0.52 |
| Phi-3 Mini-Inst | **0.83** / 0.78 | 0.80 / 0.64 | 0.82 / **0.81** | 0.81 / 0.79 | 0.82 / 0.79 | 0.74 / 0.74 | 0.58 / 0.58 |
| Mistral-7B-Inst | **0.86** / 0.84 | 0.85 / **0.85** | 0.81 / 0.80 | 0.81 / 0.81 | 0.80 / 0.77 | 0.74 / 0.58 | 0.60 / 0.50 |
| Falcon-7B Inst | **0.80** / **0.78** | 0.78 / 0.66 | **0.80** / 0.77 | **0.80** / **0.78** | 0.79 / 0.77 | 0.78 / 0.73 | 0.55 / 0.66 |

Table 5: AUC/Accuracy on **trivia** dataset.

| Model | HalluFieldSE | HalluField | KLE | SE | CE | RE | P(True) |
|---|---|---|---|---|---|---|---|
| LLaMA-2 7B | **0.83** / **0.79** | 0.81 / 0.77 | **0.83** / **0.79** | **0.83** / **0.79** | 0.82 / 0.78 | 0.77 / 0.68 | 0.52 / 0.55 |
| LLaMA-2 7B-Chat | 0.82 / **0.78** | 0.78 / 0.72 | **0.83** / 0.77 | 0.82 / 0.77 | 0.82 / 0.77 | 0.75 / 0.71 | 0.53 / 0.51 |
| LLaMA-2 13B-Chat | **0.79** / 0.72 | 0.76 / 0.69 | 0.78 / **0.75** | **0.79** / 0.74 | 0.78 / 0.73 | 0.73 / 0.68 | 0.62 / 0.66 |
| LLaMA-3.2 1B | **0.88** / 0.80 | 0.86 / **0.81** | 0.85 / 0.75 | 0.84 / 0.75 | 0.83 / 0.75 | 0.79 / 0.71 | 0.50 / 0.52 |
| LLaMA-3.2 1B-Inst | **0.85** / **0.77** | 0.82 / 0.76 | **0.85** / 0.76 | 0.84 / 0.75 | 0.83 / 0.75 | 0.78 / 0.75 | 0.50 / 0.46 |
| LLaMA-3.2 3B | 0.77 / 0.71 | **0.80** / **0.75** | 0.73 / 0.65 | 0.73 / 0.67 | 0.73 / 0.66 | 0.58 / 0.48 | 0.51 / 0.51 |
| Phi-3 Mini-Inst | 0.82 / 0.77 | **0.85** / **0.78** | 0.80 / 0.75 | 0.80 / 0.75 | 0.78 / 0.70 | 0.75 / 0.68 | 0.52 / 0.47 |
| Mistral-7B-Inst | 0.80 / **0.75** | 0.79 / 0.73 | **0.81** / **0.75** | **0.81** / **0.75** | **0.81** / **0.75** | 0.78 / 0.72 | 0.51 / 0.52 |
| Falcon-7B Inst | 0.85 / 0.79 | **0.87** / **0.81** | 0.84 / 0.75 | 0.84 / 0.78 | 0.83 / 0.75 | 0.72 / 0.70 | 0.59 / 0.45 |

Table 6: Comparison of running time (per query) and auxiliary model usage across hallucination detection methods. The running time only includes the time to process the perturbations generated from the models (not the time to generate the perturbations, which is shared among methods).

| | HalluFieldSE | HalluField | KLE | SE | CE | RE |
|---|---|---|---|---|---|---|
| Need extra LLM | Yes | No | Yes | Yes | Yes | No |
| Running time (sec) | 41.08 | $1 \times 10^{-4}$ | 41.09 | 41.08 | 41.08 | $1 \times 10^{-5}$ |

**Discussion:** The strong performance of HalluField and HalluFieldSE is threefold. First, HalluField directly leverages raw logit information, which is partially lost during semantic clustering. Second, HalluField relies on free energy and changes in potential, both of which can be computed without auxiliary LLMs. This design avoids errors introduced by additional models, particularly in cases where semantic clustering is ambiguous or difficult, as commonly observed in the *bioasq* dataset, where highly technical responses make it challenging for SE and KLE to form accurate clusters. Finally, the information captured by free energy and potential change is complementary to semantic entropy, and their integration in HalluFieldSE yields the best overall results.

**Running time:** Table 6 compares the per-query running time and auxiliary model requirements of detection methods. The results show a clear trade-off between efficiency and reliance on external LLM calls. Methods such as KLE, SE, and CE require querying an auxiliary language model, leading to substantially higher runtime (around 41 seconds per query). In contrast, HalluField avoids this dependency and achieves near-instantaneous runtime ($10^{-4}$ seconds), making it orders of magnitude faster. Since HalluFieldSE utilizes SE, it requires a similar computational cost. These results underscore that HalluField provides the best balance of computational efficiency and detection capability, while HalluFieldSE offers improved performance at the cost of additional compute time.

## 6 CONCLUSION

We introduced HalluField, a field-theoretic algorithm for hallucination detection in LLMs, grounded in a variational principle and thermodynamic intuition. By modeling the stability of energy and entropy distributions under temperature perturbations, HalluField identifies hallucinations as instabilities in the energy landscape. Experiments across multiple datasets and models show that HalluField and its variant HalluFieldSE achieve state-of-the-art detection performance while remaining computationally efficient, operating directly on logits without fine-tuning or auxiliary networks. These results demonstrate the promise of physics-inspired methods for improving the reliability of LLMs and open opportunities for extending this perspective to broader challenges in trustworthy AI.

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

## A  HEURISTIC BEHAVIORS OF THE FREE ENERGY AND THE ENTROPY

In this appendix, we present experimental results illustrating the behavior of our proposed quantity in non-hallucinated and hallucinated responses generated by LLaMA-2 7B-Chat, LLaMA-3.2 3B, and Phi-3 Mini-Inst. The experiments are conducted on the triviaQA and squad datasets.

The results are shown in Figures 4–9. Across different temperatures, models, and datasets, we consistently observe that hallucinated responses yield statistically higher values of all three measures: the base energy variation $\Delta\mathbb{B}_Q$, the change in potential $\Delta\mathbb{P}_Q$, and the change in entropy $\Delta(T\mathbb{H}_Q)$. These findings align with our hypothesis stated in Section 3.

However, we observe that the ability to distinguish hallucinated from non-hallucinated responses varies across different temperatures. This observation motivates the design of HalluField, a method that aggregates this rich information into an effective approach for hallucination detection.

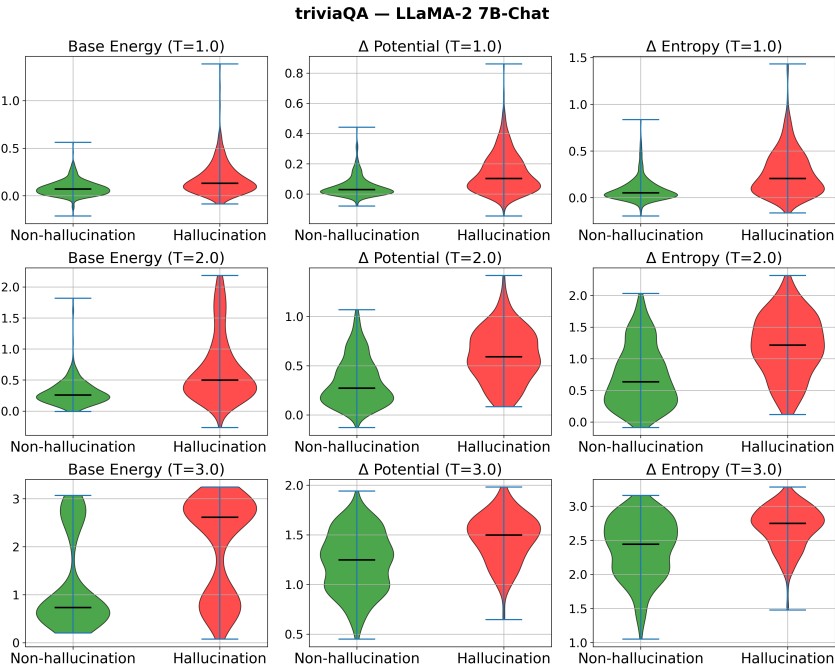

Figure 4: Behaviors of the base energy variation $\Delta\mathbb{B}_Q$, the change in potential $\Delta\mathbb{P}_Q$, and the change in entropy $\Delta(T\mathbb{H}_Q)$ between non-hallucinated and hallucinated responses of LLaMa-2-7B-Chat in triviaQA dataset

## B  VARIATIONAL PRINCIPLE ON SEQUENCES OF TOKENS

This appendix provides our formulation of the variation of a functional defined on the space of sequences of tokens; we will subsequently define in Appendix B.1 a parametrized version of the variation which allows for one to compute variations as a model quantity (e.g., temperature) varies. We then specifically consider a free energy functional (Appendix B.2) and an entropy functional (Appendix B.3) on the space of sequences of tokens.

Let $\mathbb{T}$ denote the space of tokens and $\mathbb{T}^*$ denote the space of sequences of tokens. Let $A_Q : \mathbb{T} \to \mathbb{R}$, referred to as a *density*; a corresponding functional $\mathbb{A}_Q : \mathbb{T}^* \to \mathbb{R}$ can be defined by summing the density over the token sequence, i.e.,

$$\mathbb{A}_Q[\boldsymbol{\tau}] = \sum_{i=1}^{N} A_Q[\tau_i]\Delta_i, \tag{18}$$

Here, $\boldsymbol{\tau}$ is a sequence of tokens of length $N$, $\Delta_i > 0$ is the distance between token depths $i$ and $i+1$, and the subscript $Q$ throughout denotes dependence on the query $Q$. Given another sequence

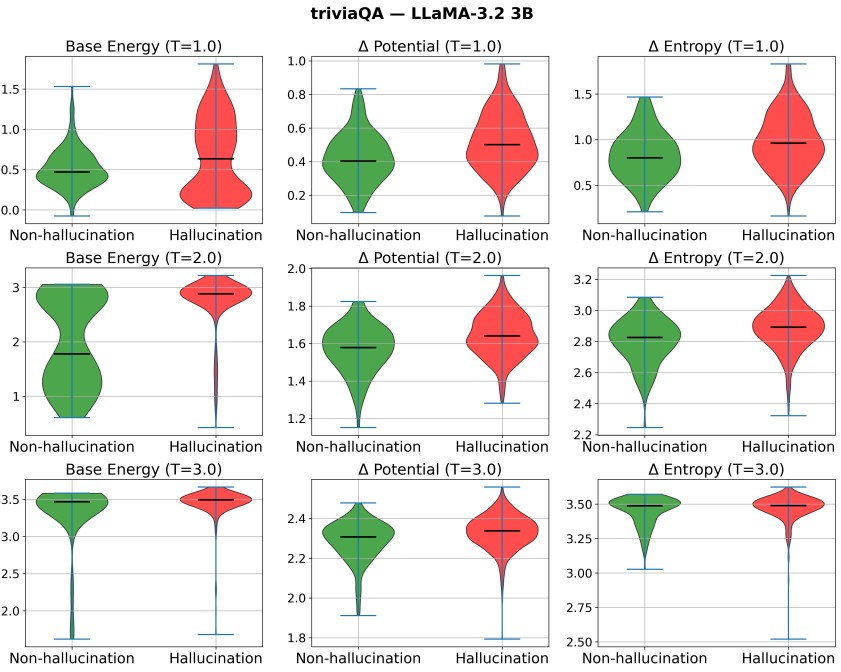

Figure 5: Behaviors of the base energy variation $\Delta\mathbb{B}_Q$, the change in potential $\Delta\mathbb{P}_Q$, and the change in entropy $\Delta(T\mathbb{H}_Q)$ between non-hallucinated and hallucinated responses of LLaMa-3.2-3B in triviaQA dataset

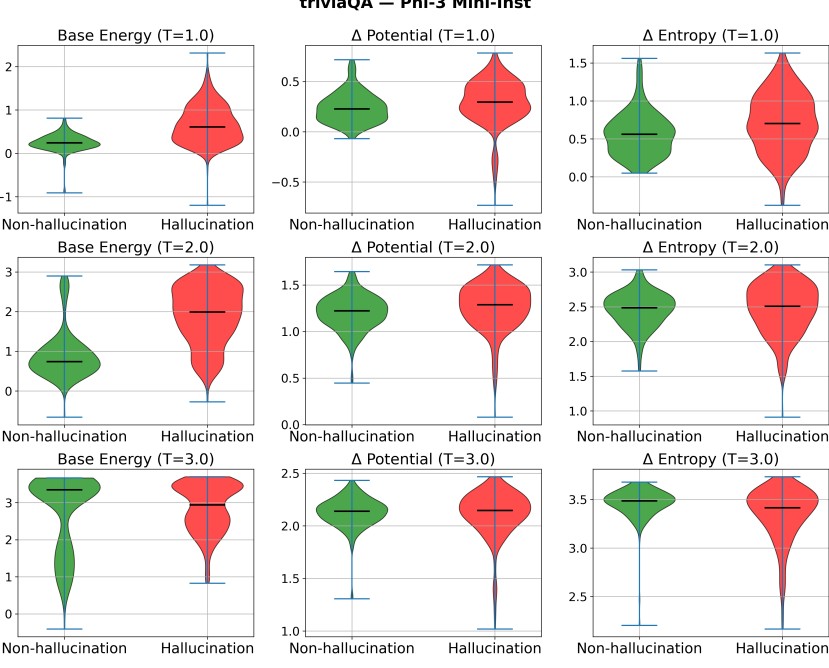

Figure 6: Behaviors of the base energy variation $\Delta\mathbb{B}_Q$, the change in potential $\Delta\mathbb{P}_Q$, and the change in entropy $\Delta(T\mathbb{H}_Q)$ between non-hallucinated and hallucinated responses of Phi-3 Mini-Inst in triviaQA dataset

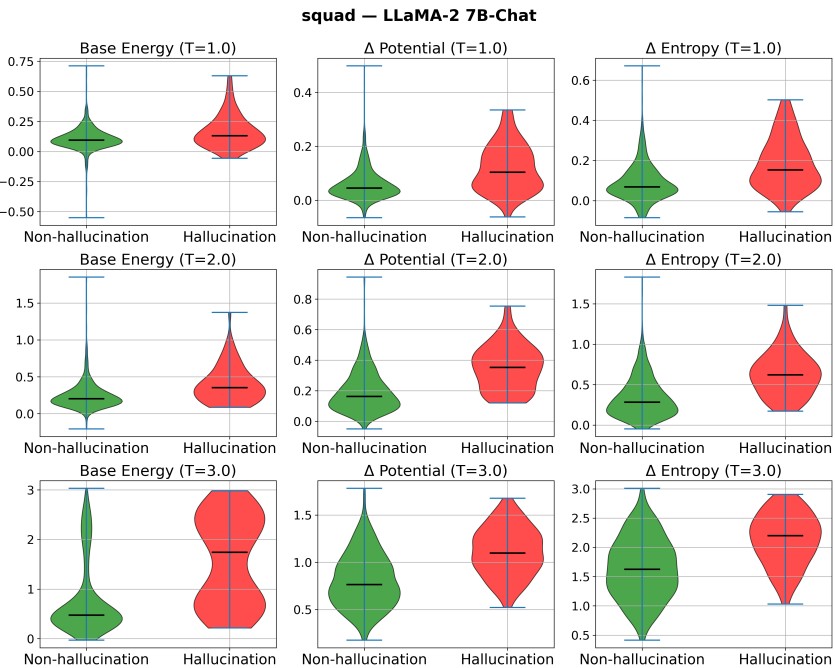

Figure 7: Behaviors of the base energy variation $\Delta\mathbb{B}_Q$, the change in potential $\Delta\mathbb{P}_Q$, and the change in entropy $\Delta(T\mathbb{H}_Q)$ between non-hallucinated and hallucinated responses of LLaMa-2-7B-Chat in squad dataset

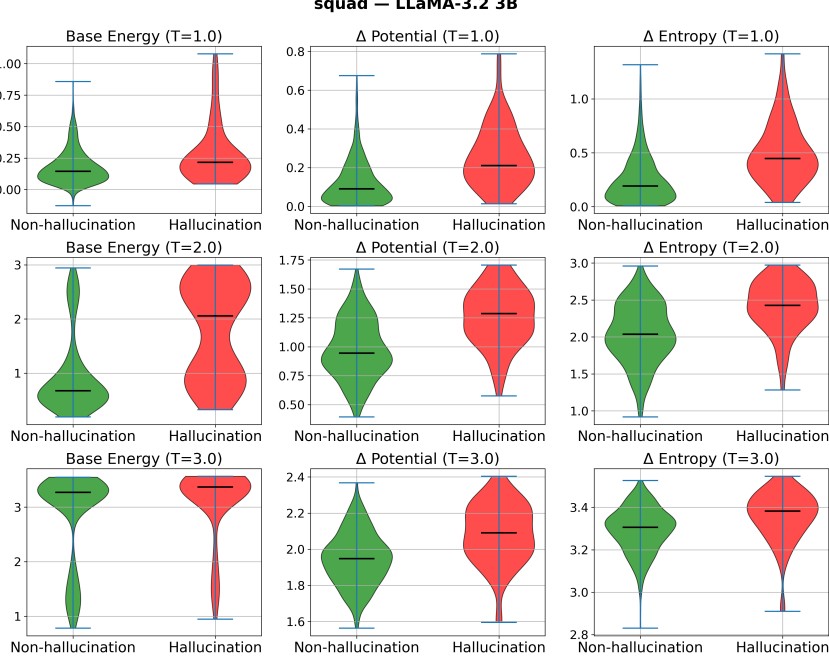

Figure 8: Behaviors of the base energy variation $\Delta\mathbb{B}_Q$, the change in potential $\Delta\mathbb{P}_Q$, and the change in entropy $\Delta(T\mathbb{H}_Q)$ between non-hallucinated and hallucinated responses of LLaMa-3.2-3B in squad dataset

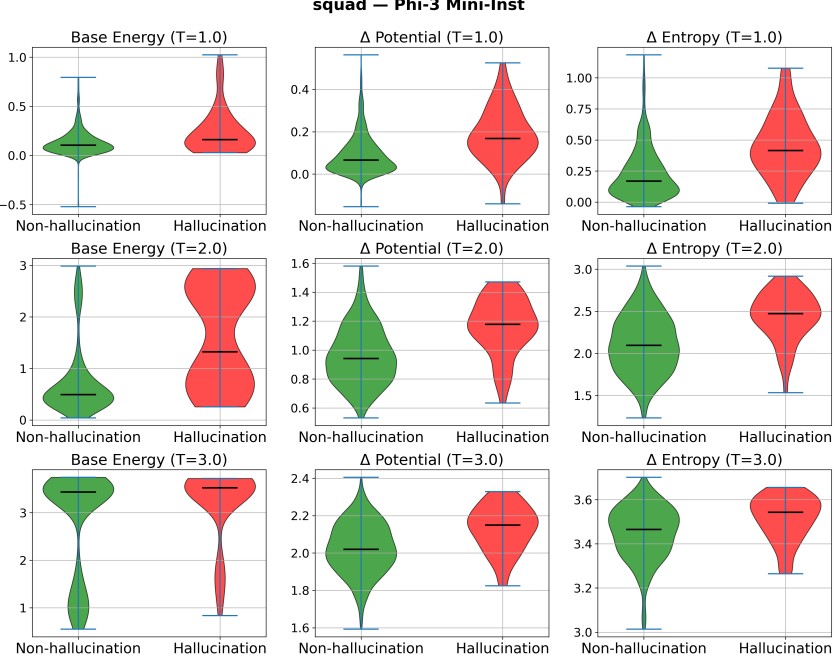

Figure 9: Behaviors of the base energy variation $\Delta\mathbb{B}_Q$, the change in potential $\Delta\mathbb{P}_Q$, and the change in entropy $\Delta(T\mathbb{H}_Q)$ between non-hallucinated and hallucinated responses of Phi-3 Mini-Inst in squad dataset

of tokens $\boldsymbol{\chi} = \{\chi_i\}_{i=1}^N$, we define the discrete variation of this functional from $\boldsymbol{\tau}$ to $\boldsymbol{\chi}$ by

$$\Delta\mathbb{A}_Q[\boldsymbol{\tau};\boldsymbol{\chi}] = \sum_{i=1}^N \frac{A_Q(\tau_i) - A_Q(\chi_i)}{d(\tau_i, \chi_i)}\Delta_i, \tag{19}$$

where $d : \mathbb{T} \times \mathbb{T} \to \mathbb{R}$ is a pseudo-distance function between tokens $\tau$ and $\chi$ which is symmetric and non-negative; in order to avoid vanishing divisors in the discrete setting, we impose that the pseudo-distance function is bounded below by $\epsilon > 0$, i.e., $d(\tau, \chi) \geq \epsilon$ for all tokens $\tau, \chi$, and the pseudo-non-degeneracy condition $d(\tau, \chi) = \epsilon$ if and only if $\tau = \chi$.

### B.1 PARAMETRIZED VARIATIONS

A simple way to compute the variation of a functional is to parametrize the domain of the functional by parameters and look at variations induced by changing those parameters. Furthermore, such parametrized variations allow one to compute how a functional changes with respect to a particular parameter; for example, the HalluField algorithm uses variations with respect to temperature.

Starting with the continuous case, let us consider a functional $\mathbb{A}$ on a space of curves,

$$\mathbb{A}[\mathbf{c}(\mathbf{r}, T)] = \int A(\mathbf{c}(\mathbf{r}, T))\, ds, \tag{20}$$

where now the curve $\mathbf{c}$ is parametrized by two parameters $\mathbf{r}$ and $T$ (in the discrete token setting, this will later be the likelihood depth and the temperature). Note that the parameters (particularly $\mathbf{r}$) may generally depend on the current time $s$ along the path.

Instead of considering generic variations of the curve $\mathbf{c}$, parametrized variations consider only variations of the curve $\mathbf{c}$ induced by varying the parameters. Particularly, the parametrized variation of this functional with respect to $T$ is

$$\Delta\mathbb{A}[\mathbf{c}(\mathbf{r}, T); \Delta T] = \int \left(\frac{\partial A}{\partial \mathbf{c}} \cdot \frac{\partial \mathbf{c}}{\partial T}\Delta T\right) ds. \tag{21}$$

Now, on to the setting of tokens. Given any two sequences, we can assume that they are the same length by trivially extending the shorter sequence. We consider a sequence of tokens parametrized by the temperature $T \in \mathbb{R}^{\geq}$, and a vector of likelihood depths $\mathbf{r} \in (\mathbb{Z}^+)^N$, where the $i^{th}$ component of this vector $r_i$ corresponds to taking the $r_i^{th}$ most likely token for the $i^{th}$ token in the sequence. We denote this token sequence by

$$\boldsymbol{\tau}(\mathbf{r}, T) \in \mathbb{T}^*. \tag{22}$$

The $i^{th}$ token of this sequence is denoted $\tau_i(r_i, T)$. Consider again a functional on the space of sequences of tokens, $\mathbb{A}_Q : \mathbb{T}^* \to \mathbb{R}$, we define the parametrized functional

$$\mathbb{A}_Q[\boldsymbol{\tau}(\mathbf{r}, T)] := \sum_{i=1}^N A_Q(\tau_i(r_i, T))\Delta_i. \tag{23}$$

Of course, the functional $\mathbb{A}_Q[\boldsymbol{\tau}(\mathbf{r}, T)]$ depends on the parameters $\mathbf{r}$ and $T$; when it is clear in context, we will simply express this as $\mathbb{A}_Q[\boldsymbol{\tau}]$. In analogy to the discrete variation equation 19 and continuous parametrized variation equation 21, we define the *parametrized discrete variation* with respect to a temperature perturbation $T \to T + \Delta T$ (where $\Delta T \geq -T$) by

$$\Delta\mathbb{A}_Q[\boldsymbol{\tau}; \Delta T] := \sum_{i=1}^N \frac{A_Q(\tau_i(r_i, T + \Delta T)) - A_Q(\tau_i(r_i, T))}{d(\tau_i(r_i, T + \Delta T), \tau_i(r_i, T))}|\Delta T|\Delta_i. \tag{24}$$

Based on this notation, we refer to the model's response that we wish to classify as either a hallucination or a non-hallucination as the *base response*. This corresponds to the path $\boldsymbol{\tau}$, parameterized by the token-likelihood choice $\mathbf{r}^0$ at the model's normal operating temperature $T^0$. In other words, the response under evaluation for hallucination can be denoted as $\boldsymbol{\tau}(\mathbf{r}^0, T^0)$. Note that under this notation, while $\boldsymbol{\tau}(\mathbf{r}^0, T)$ denotes a different path, it corresponds to the same sequence of tokens in the base response.

Given a query $Q$, base temperature $T^0$, and the corresponding response likelihood path $\boldsymbol{\tau}$, HalluField considers a uniform distance between token depths $\Delta_i = 1/N$ and uses the following parametrized discrete variation with respect to the temperature:

$$\Delta\mathbb{A}_Q[\boldsymbol{\tau}; \Delta T] = \frac{1}{N}\sum_{i=1}^N |\Delta T|\frac{A_Q(\tau_i(r_i^0, T^0 + \Delta T)) - A_Q(\tau_i(r_i^0, T^0))}{d(\tau_i(r_i^0, T^0 + \Delta T), \tau_i(r_i^0, T^0))}. \tag{25}$$

Another advantage of parametrizing the variations is that it eliminates the need to compute distances directly between tokens (which could alternatively be computed using embeddings, but at a significantly higher computational cost). Instead, we only require a distance function defined over tokens that differ with respect to the chosen parameter—in this case, the temperature $T$. A natural and simplest choice for such a distance is then

$$d(\tau(r_i^0, T + \Delta T), \tau(r_i^0, T)) = |\Delta T|. \tag{26}$$

This gives the following simplification of the parametrized discrete variation

$$\Delta\mathbb{A}_Q[\boldsymbol{\tau}; \Delta T] = \mathbb{A}_Q[\boldsymbol{\tau}(\mathbf{r}^0, T^0 + \Delta T)] - \mathbb{A}_Q[\boldsymbol{\tau}(\mathbf{r}^0, T^0)]. \tag{27}$$

**Total variation.** Now, we define the total variation of the functional $\delta\mathbb{A}_Q$ by a weighted sum of the parametrized discrete variations $\Delta\mathbb{A}_Q[\boldsymbol{\tau}; \Delta T]$ for various choices of $\Delta T = \Delta T_1, \ldots, \Delta T_n$. That is,

$$\delta\mathbb{A}_Q := \sum_{\Delta T = \Delta T_1}^{\Delta T_n} w(T, \Delta T)\Delta\mathbb{A}_Q[\boldsymbol{\tau}; \Delta T].$$

Intuitively, the total variation is a linear combination of several individual variations; the choice of a linear combination comes from the fact that variations measure the linear response of a functional to a perturbation in the parameter.

## B.2   THE FREE ENERGY FUNCTIONAL

Given a query $Q$, we aim to define a scalar quantity that captures the essence of a token sequence just like energy captures the essence of a trajectory in classical physics. We want this scalar property to be additive and dependent on the length of the token sequence for a fair comparison between $\tau$ of different lengths and for satisfying linearity defined in equation 23. In thermodynamics, such a quantity is called an extensive property. We call this the *free energy* $F_Q : \mathbb{T}^* \to \mathbb{R}$ and define it as a map such that $F_Q$ is continuous, monotonic, and a thermodynamically extensive function of the token sequence. We derive the functional form of this energy from the following statistical arguments (see also Landau & Lifshitz (1980)). Let the probability of getting a token sequence $\tau$ from query $Q$ be given by the conditional probability $P(\tau|Q)$. For the $i^{th}$ token $\tau_i$ in the sequence $\tau = \{\tau_i\}_{i=1}^N$, its probability is conditioned on all previous tokens, $P(\tau_i|\{\tau_j\}_j^{i-1}, Q)$. Consequently, the following relation holds between the joint probability $P(\tau, Q)$ and conditional probabilities of all tokens:

$$P(\tau, Q) = P(\tau_1, \tau_2, ..., \tau_N, Q) = \prod_i^N P(\tau_i|\{\tau_j\}_{j=1}^{i-1}, Q)P(Q) = P(\tau|Q)P(Q).$$

Due to the extensive property, a free energy defined on the sequence of tokens must be a function of these conditional probabilities (Landau & Lifshitz, 1980):

$$\mathbb{F}_Q(\tau|Q) = F_Q\left(\prod_i^N P(\tau_i|\{\tau_j\}_{j=1}^{i-1}, Q)\right) = \sum_i^N F_Q(P(\tau_i|\{\tau_j\}_{j=1}^{i-1}, Q)). \tag{28}$$

Equation 28 is a logarithmic functional equation, which leads to the following family of possible functions via Cauchy's functional equation:

$$F_Q(\tau) = -k \log P(\tau|Q). = -k \log P(\tau|Q).$$
$$F_Q(\tau_i) = -k \log P(\tau_i|\{\tau_j\}_{j=1}^{i-1}, Q) = -\log P(\tau_i|\{\tau_j\}_{j=1}^{i-1}, Q), \tag{29}$$

where $k$ is a positive real number. As convention, we take $k = 1$.

Its variation is then given by

$$\Delta\mathbb{F}_Q[\tau; \Delta T] := \sum_{i=1}^N \frac{F_Q(\tau_i(r_i, T + \Delta T)) - F_Q(\tau_i(r_i, T))}{d(\tau_i(r_i, T + \Delta T), \tau_i(r_i, T))}|\Delta T|\Delta_i. \tag{30}$$

To explain why the absolute value $|\Delta T|$ appears, let us check the intuition for the sum appearing in equation 24. Consider a perturbation in the temperature. For $\Delta T > 0$, $F_Q(\tau_i(r_i, T + \Delta T)) > F_Q(\tau_i(r_i, T))$ (since the sequence of tokens generated at lower temperature should have higher probability and hence, lower free energy) which makes the sum positive, i.e., we have a positive variation in the free energy functional. On the other hand, if $\Delta T < 0$, $F_Q(\tau_i(r_i, T + \Delta T)) < F_Q(\tau_i(r_i, T))$, leading to a negative variation in the free energy functional. Mathematically, we do not keep track of the sign of the factors of $\Delta T$ appearing in the summands of equation 24 since this is already accounted for in the difference of the free energies. Furthermore, $|\Delta T|$ is the unsigned measure of the interval $[T, T + \Delta T]$.

Note that the computation of $\Delta\mathbb{F}_Q[\tau; \Delta T]$ in equation 30 requires to compute $F_Q(\tau_i(r_i, T+\Delta T)) - F_Q(\tau_i(r_i, T))$, which is the difference between the free energy of two exactly generated sequences of tokens but at different temperatures. In practice, this can be challenging since, at a large $\Delta T$, the LLM will become too random for us to observe the same response at a lower temperature. We discuss how to approximate this quantity in the description of our algorithm in Section 4.

## B.3   THE ENTROPY FUNCTIONAL

Let us return to the continuous case. To define the entropy functional, we consider a family of curves given by varying $\mathbf{r}$. Namely, for the curve $\mathbf{c}(\mathbf{r}, T)$, for each time $s$, we take the parameter $r = \mathbf{r}(s)$ to be distributed by some probability distribution $p(\mathbf{c}(r, T)(s))$. The entropy of this family of curves at time $s$ is defined to be

$$H(\mathbf{c}(\cdot, T)(s)) = -\sum_r p(\mathbf{c}(r, T)(s)) \log p(\mathbf{c}(r, T)(s)),$$

where the sum is over all possible states parametrized by $r$. The *entropy functional* is given by integrating the entropy over all times $s$, i.e.,

$$\mathbb{H}(\mathbf{c}(\cdot, T)) = \int H(\mathbf{c}(\cdot, T)(s)) ds,$$

which is interpreted as the total entropy of this family of parametrized curves.

Analogous to our discussion of the free energy, we can define a discrete analogue of the entropy functional on sequences of tokens using the Shannon entropy,

$$\mathbb{H}_Q(\boldsymbol{\tau}(\cdot, T)) := \sum_{i=1}^{N} H_Q(\tau_i(\cdot, T))\Delta_i, \tag{31a}$$

$$H_Q(\tau_i(\cdot, T)) = -\sum_{r=1}^{|\mathbb{T}|} P(\tau_i(r, T)|\{\tau_j\}_{j=1}^{i-1}, Q) \tag{31b}$$
$$\times \log P(\tau_i(r, T)|\{\tau_j\}_{j=1}^{i-1}, Q).$$

The definition of the entropy functional, equation 31a and equation 31b, can be interpreted as a discrete double integral over the length of the token sequence in one direction and over all possible likelihood depths in the other direction (where the likelihood depths run from 1 to the total context length $|\mathbb{T}|$).

According to equation 6, we are interested in the parametrized discrete variation of the temperature-entropy functional (which is simply the product of the temperature and the entropy functional). Proceeding similarly to the free energy functional, the parametrized discrete variation of the temperature-entropy functional, with respect to a temperature perturbation $T \to T + \Delta T$, is given by

$$\Delta(T\mathbb{H}_Q)[\boldsymbol{\tau}; \Delta T] := \sum_{i=1}^{N} \frac{(T + \Delta T)H(\tau_i(\cdot, T + \Delta T)) - TH(\tau_i(\cdot, T))}{d(\tau_i(\cdot, T + \Delta T), \tau_i(\cdot, T))} |\Delta T| \Delta_i. \tag{32}$$

These parametrized discrete variations of the free energy functional, $\Delta \mathbb{F}_Q$, and of the temperature-entropy functional, $\Delta(T\mathbb{H})$, form the basis of the HalluField algorithm, which considers a weighted combination of these variations to compute the total variation

$$\delta \mathbb{U}_Q = \delta \mathbb{F}_Q + \delta(T\mathbb{H}_Q), \tag{33}$$

as described in Section 3.

## C  EXPERIMENTAL SETTINGS

Our experiments ran on a cluster whose nodes each comprised an AMD EPYC 7713 (Rome) 64-core, 2 GHz CPU, 256 GB system memory, and 4× NVIDIA A100 Tensor Core GPUs (40 GB HBM per GPU). We used the GPUs solely to compute model logits, while the subsequent post-processing operations were executed on the CPUs due to their lightweight nature.

For each dataset, we follow the evaluation protocol of (Farquhar et al., 2024; Nikitin et al., 2024) and assess hallucination rates on 500 samples. For HalluField, we generate 50 perturbations per temperature. The benchmark methods use a comparable number of perturbations, namely $50 \times$ Number of temperatures. We typically use the temperature set $\{1.0, 1.5, 2.0\}$. For models with different temperature scaling, such as LLaMa-2-7B-Chat, we instead use $\{1.0, 2.0, 3.0\}$. For KLE (Nikitin et al., 2024), which integrates multiple parameters and kernel methods, we adopt the strongest reported variant, $KLE_{Heat}$, as recommended by the authors. We further fine-tune its parameters and set $t_{KLE} = 0.2$ and $\alpha_{KLE} = 0.5$.

At high temperatures, models tend to generate longer responses, which substantially increases evaluation time since each output must be processed by another LLM to obtain ground-truth labels (Farquhar et al., 2024). Following prior work, we cap the number of generated tokens at 50 to control runtime. For the accuracy results, the cutoff points for all methods are determined using the Youden index optimal criterion (Fluss et al., 2005).

