# OpenReview forum: "HalluField: Detecting LLM Hallucinations via Field-Theoretic Modeling"
_ICLR.cc/2026/Conference — ICLR 2026 Conference Withdrawn Submission_

### Official Review · Reviewer_nx7L · 2025-10-29

**Soundness:** 3
**Presentation:** 3
**Contribution:** 2
**Rating:** 4
**Confidence:** 4

**Summary:**

The paper proposes a method to detect hallucinations in LLM responses by using semantic instability to determine the likelihood of hallucinations. The authors do this by measuring the change in internal energy of the tokens, inspired by laws of thermodynamics. They add to the extensive literature on uncertainty-based methods for hallucination detection, and test their method on four QA benchmarks across several LLMs, in comparison to other hallucination detection baselines.

**Strengths:**

-The time complexity of the authors’ approach is much more efficient than the SOTA baselines.
-Four different LLM-families are used to evaluate HalluField on four datasets, which is inconsistent with other experimental set-ups in the literature.

**Weaknesses:**

- Several more recent hallucination-detection baselines are missing: SINdex (Abdaljalil et al., 2025) and RACE (Wang et al., 2025).
- HalluField does not outperform other baselines in all contexts, as shown in the results tables (2-4)
- Only open-source LLMs can be used for this method. Is there  a way to adapt this to black-box LLMs as well?

**Questions:**

- Why is it that the performance is the most inconsistent for the TriviaQA dataset?
- See Weaknesses for further questions.

---

### Official Review · Reviewer_MqLb · 2025-10-30

**Soundness:** 2
**Presentation:** 2
**Contribution:** 2
**Rating:** 4
**Confidence:** 4

**Summary:**

Goal:
- This paper addresses the problem of hallucination detection, e.g., estimating how likely a model’s generated answer is correct or factually consistent. The goal is to produce a numerical score that reflects the reliability of each model response, with higher scores for factual answers and lower scores for hallucinated ones.

Method:
- The proposed method builds on the line of research represented by Semantic Entropy (SE), which estimates uncertainty by sampling multiple responses and comparing their semantic variability. However, instead of relying on a separate semantic clustering step as in SE, this work directly analyzes the model’s internal token-level probabilities under different temperature settings. By quantifying how these probabilities fluctuate with temperature, the method captures the model’s intrinsic response stability.

Contribution
- The main contribution is the new hallucination detection algorithm, HalluField, which measures uncertainty without any semantic clustering or auxiliary models. It leverages the variation of token probabilities across temperature perturbations to derive an uncertainty score that distinguishes factual from hallucinated outputs.

**Strengths:**

1. The proposed algorithm itself is new and novel, without requiring an additional semantic clustering model.
2. The evaluation metrics and datasets follow the standard literature.

**Weaknesses:**

1. The paper’s exposition could be clearer. Introducing the first law of thermodynamics as an analogy does not necessarily make the story more compelling, especially when there is no theoretical grounding; instead, it adds unnecessary conceptual and notational complexity. The notation is rather dense and sometimes unconventional, making a relatively simple idea appear overly complicated. Readers need to spend substantial effort cross-referencing earlier equations to fully understand the computation process. For instance, Algorithm 1, which references seven equations from previous sections, is difficult to follow and fails to serve its intended role as a step-by-step illustration of the method.
2. The experimental evaluation relies on a relatively narrow selection of benchmarks. It would be good to see how the method performs across more diverse or realistic application domains. Besides, what kind of prompt do you use for the experiments? Is it naive short-form QA, or does it also include COT reasoning?
4. Only 1 single-sample-based method P(true) is considered.
3. Regarding the evaluation pipeline, to what extent do you think the observed AUC improvements genuinely reflect better hallucination detection comparison between different models, since the accuracy and output generations for each compared method are also changing?

**Questions:**

1. Beyond the algorithmic formulation itself, what is the main conceptual or methodological takeaway you hope the audience will gain from this work? In other words, what broader insight or principle about the hallucination detection task and experiments do you think this paper will bring to this community?
2. Have you considered conducting an ablation study to analyze how each component of your method contributes to the overall performance? It would be helpful to understand which part of the design is most critical for improving accuracy or stability.
3. In the abstract, you describe the proposed approach as a general-purpose hallucination detection algorithm. Could you elaborate on how this method could be applied to more realistic or diverse application scenarios? Additionally, more insights on what kind of performance we should expect in such settings, and why this is the case, are appreciated.

---

### Official Review · Reviewer_pMYv · 2025-10-31

**Soundness:** 2
**Presentation:** 2
**Contribution:** 2
**Rating:** 4
**Confidence:** 3

**Summary:**

The paper proposes a novel hallucination detection framework inspired by thermodynamics. The authors model the generation process of large language models (LLMs) as an analogue to a thermodynamic system, introducing quantities such as free energy and temperature–entropy to describe token-level behaviors. The central algorithm, HalluField, estimates the "internal energy variation" of a sequence by aggregating two key terms: (1) the variation of token-level negative log-likelihood ("free energy") and (2) the variation of next-token entropy across multiple temperature perturbations. Empirically, HalluField is significantly faster because it does not require auxiliary LLM queries. Its variant, HalluFieldSE, achieves modest but consistent AUC gains over prior entropy-based detectors (e.g., Semantic Entropy and Kernel Language Entropy) across multiple benchmarks.

While the proposed approach successfully identifies temperature-dependent stability patterns that correlate with hallucination likelihood, the paper’s extensive thermodynamic analogy is largely explanatory in nature. The empirical contributions could be presented and motivated without invoking physical conservation laws. Nonetheless, the paper presents interesting empirical findings about the sensitivity of token probability distributions under temperature perturbation and offers a practical and efficient detection scheme.

**Strengths:**

1. *Empirical novelty*: The discovery that cross-temperature instability of token probabilities and entropies correlates with hallucination is an interesting and empirically supported observation.
1. *Practical efficiency*: HalluField requires only base-model logits and is orders of magnitude faster than detectors relying on auxiliary LLMs.
2. *Clarity of algorithmic pipeline*: Algorithm 1 and equations (13–17) describe a clear procedure.
3. *Cross-dataset evaluation*: The authors test across multiple LLMs and QA datasets, providing a reasonably broad empirical view.
4. *Complementarity*: Combined with Semantic Entropy (HalluFieldSE), the method often yields incremental performance improvements.

**Weaknesses:**

1. *Unnecessary and weakly supported analogy*: The thermodynamic framework adds complexity without theoretical necessity. It lacks formal justification or derivation from model behavior. Token-level negative log-likelihood and entropy can be motivated directly from information theory and prior NLP literature.
2. *Lack of statistical rigor*: Reported AUC gains (0.01–0.09) lack confidence intervals or significance tests, making it unclear whether the improvements are statistically meaningful.
3. *Missing ablations*: No experiments isolate the contribution of each component, such as removing the temperature–entropy term, testing a single temperature, or varying the weighting scheme.
4. *Overreliance on appendix*: Crucial figures (e.g., cross-temperature separation plots) and accuracy metrics are in the supplementary material. It undermines the completeness of the main paper.

**Questions:**

1. Can you provide statistical confidence intervals or significance tests for the AUC improvements to demonstrate robustness?
2. How sensitive are the results to the temperature schedule and weighting functions in Eq. (14)? Have you tested alternative schedules?
3. Could you include ablations removing one component at a time (Δ𝐹 only, Δ(𝑇𝐻) only, or single temperature) to confirm the contribution of each term?
4. For practical deployment, is there a recommended or fixed threshold (cutoff) for hallucination detection across models, or is threshold calibration required for each dataset?

---

### Note · Authors · 2025-11-20

**Comment:**

After considering the reviews, we have decided to withdraw the submission to allow for further improvement.

**Withdrawal Confirmation:**

I have read and agree with the venue's withdrawal policy on behalf of myself and my co-authors.